# Motherhood in Alternative Detention Conditions: A Preliminary Case-Control Study

**DOI:** 10.3390/ijerph19106000

**Published:** 2022-05-15

**Authors:** Carlo Lai, Linda Elisabetta Rossi, Federica Scicchitano, Chiara Ciacchella, Mariarita Valentini, Giovanna Longo, Emanuele Caroppo

**Affiliations:** 1Department of Dynamic and Clinical Psychology, and Health Studies, Sapienza University, 00185 Rome, Italy; rossi.1899893@studenti.uniroma1.it (L.E.R.); scicchitano.1813358@studenti.uniroma1.it (F.S.); chiara.ciacchella@uniroma1.it (C.C.); 2Research Center Health Human Care and Social Intercultural Assessment-He.R.A, Università Cattolica del Sacro Cuore, 00168 Rome, Italy; valentini@satiscentroclinico.com; 3Satis-Centre for Clinical Psychology, 00198 Rome, Italy; 4Associazione A Roma, Insieme Leda Colombini, 00186 Rome, Italy; aromainsieme@gmail.com; 5Department of Mental Health, Local Health Unit Roma 2, 00159 Rome, Italy; emanuelecaroppo@gmail.com

**Keywords:** imprisoned mothers with children, alternative detention conditions, Group Homes, parenting stress, child behavior, maternal attachment

## Abstract

Many women in detention are mothers and often the sole caregivers of their children. Italy, as most European countries, allows mothers to keep their children with them in detention, with the aim of preserving the fundamental bond between mother and child. Since prison does not seem to provide a good environment for the child’s growth, there are different alternative residential solutions, such as Group Homes. The aim of this preliminary study was to explore the differences between mothers living in detention through alternative measures with their children and mothers who are not detained regarding parenting stress, child behavior from the parent’s perspective, and maternal attachment. Twelve mothers were enrolled in this study, divided equally between the detained and the control groups. Both groups’ participants completed a three-questionnaire battery in order to assess parenting stress, child’s behavior, and maternal attachment. The analyses of variance showed significant differences between the two groups, with the detained group reporting higher scores than the control group in almost all the subscales of parenting stress. The results highlighted that imprisoned mothers might experience more stress than the general population. There is a need to design intervention programs to support parenting in detention.

## 1. Introduction

Although, as of today, the female population represents a small portion of the total number of incarcerated people in Europe and across the world, this is increasing every year [1]. Many women in detention are mothers and find themselves as the sole caregiver of their children. In most European countries, it is possible for young children to spend the first years of their lives with their mothers, either in prison or in alternative places of detention [2,3].

In Italy, as in most European countries, Law No. 354 of 26 July 1975 allows incarcerated mothers to keep their children with them, under special circumstances, until the age of three. The aim is to preserve the dyadic mother-child fundamental bond from birth [4,5] as both child psychiatrists and other specialists in the international scientific community agree that mothers and children should not be separated in the early years of the children’s lives [1,6,7]. However, despite the efforts to make prison nursery sections livable for children, these remain environments that do not match the needs for socialization and psycho-physical development that growth requires [8].

Given these opinions, there are data resulting from different studies to consider.

First, it seems reasonable to expect mothers in detention to feel powerless, to experience distress regarding their role, and stress related to deprived conditions [9,10,11].

Second, the negative outcomes for children associated with parental imprisonment have been well documented by researchers [1,12]. Children whose parents are involved in the criminal justice system in the UK, USA, and Canada, are at increased risk of (1) developing behavioral difficulties (e.g., engaging in delinquent behavior) and (2) reporting higher levels of externalizing problems, which can result in aggressive and antisocial behaviors [12,13,14,15]. Moreover, in Italy, it has been reported that children who live in detention with their mothers frequently suffer from sleep disorders, inappetence, apathy, and restlessness [8].

In addition, multiple studies conducted in the US have found that the reduced contact between mother and child during imprisonment may exacerbate attachment disorders and damage the developing attachment bond [7,16,17]. According to Borelli and colleagues [18], it is possible that, compared to preoccupied attachment, both secure and dismissing attachment are more suitable for mothers in detention, but only in terms of perception of social support, parenting competency, depressive symptoms, and substance abuse history [18]. Finally, there is a significant gap in the literature on mothers’ attachment to their children during cohabitation in detention, especially in Italy.

All consequences and effects cited above led the Italian Legislative system to reconsider accommodations for imprisoned mothers. Thus, in 2001, Law n. 40 made it possible for imprisoned mothers to serve their sentences through three alternative measures of detention, allocated in appropriate facilities [19]. The first alternative measure is a prison reserved for mothers who reside with children and which in Italy is known as “Istituti a Custodia Attenuata per detenute Madri” (ICAM). This alternative measure is a prison for all intents and purposes except for the establishment of environmental modifications to prevent children from recognizing the prison setting in which they are housed in [20]. The second alternative measure is home detention, applied for minor crimes, which allows the women to spend the sentence at their own home with their children, under electronic and police monitoring. When home detention is not possible due to adverse housing conditions, a third alternative measure is applied, the protected Group Homes. This kind of accommodation consists of spacious houses where some dyads of mothers and children can have their own room and space. While there are several controls carried by the police [21], the living environment more closely resembles that of a domestic home. Unfortunately, in Italy, the availability of protected Group Homes is limited by the lack of funds provided by the State [22], and there are only two existing Group Homes: the “Casa C.I.A.O.” in Milan and the “Casa di Leda Colombini” in Rome [23].

Although alternative solutions to prison exist, in the literature, there is a lack of data regarding their possible effects on the quality of mother–child relationships. It appears to be important to provide preliminary evidence about the impact of alternative measures of detention by comparing a group of women in detention through alternative measures with a group of mothers who are not detained. Therefore, the aim of this preliminary study was to explore the differences between mothers living in detention through alternative measures with their children and mothers who are not detained regarding parenting stress, child behavior from the parent’s perspective, and maternal attachment. Our first hypothesis was that mothers currently detained through alternative measures would experience higher levels of stress compared to mothers not in detention. A second hypothesis was that children living with mothers detained in alternative measures facilities would report higher levels of difficult behavior because of the delicate living conditions. The third hypothesis was that maternal attachment would be higher in women detained in alternative measures facilities, presumably because of a greater protective instinct.

## 2. Materials and Methods

### 2.1. Participants

This study was approved by the Ethics Committee of the Department of Dynamic and Clinical Psychology, and Health Studies of the Sapienza University of Rome.

After reading and signing the Informed Consensus, 12 mothers (*M* = 35.0; *SD* = 7.6 years old) were recruited.

Participants belonged to two different groups: six in the detained and six in the control group.

The detained group’s inclusion criteria were the mother’s age of >18 years, a child living with her at least for the last 2 years, the child’s age of 2–12 years, or occupancy in an alternative measure scenario, such as in a Group Home or home detention location. Group Homes are spacious houses, often confiscated from organized crime organizations, where mothers and children reside together in their own rooms. They have facilities that can be commonly found in a regular living environment, e.g., kitchen, bathroom, living room, and outside garden. Detention in Group Homes is applied when the mother’s original home conditions are not in line with standard hygiene guidelines. Home detention is employed, instead, when the mother’s home is considered a safe place for her and her child; therefore, she is offered the chance to serve her sentence there, together with her child, with a precise timeline of controls enforced by police officers. The control group’s inclusion criteria were a lack of previous convictions or prison residency, the mother’s age of >18 years, a child living with her, and the child’s age of 2–12 years.

The exclusion criterion for both groups was an inability to speak and read Italian.

The recruitment of the women in Group Home detention facilities took place at the “Leda Colombini” Group Home in Rome. The recruitment of the women in home detention scenarios took place at the head office of the “A Roma, insieme Leda Colombini” Association and at the Roma camp of Castel Romano, which is a nomad camp located in Rome. Members of the Roma and Sinti communities are allowed to spend their house arrest inside the nomad camp.

The recruitment of the control group took place in their private homes.

### 2.2. Materials

Three different self-report questionnaires were administered to assess the variables of interest.

The first questionnaire administered was the Parenting Stress Index Short Form (PSI-SF) [24], a questionnaire consisting of 36 items and four main scales. Participants were asked to rate their responses on a five-point Likert scale (from strongly agree to strongly disagree) to the extent to which they would agree with each sentence. The Parental Distress Scale assesses the levels of personal distress associated with factors such as depression, couple conflict, or the necessity of the parents to adapt to the needs of a child. Subsequently, the Parent–Child Dysfunctional Interaction Scale assesses the level of dissatisfaction the parent feels about his or her interactions with the child and his or her disapproval of the child’s behavior. Finally, the Difficult Child Scale assesses a parent’s perception of a child’s capability for emotional regulation. The PSI-SF also comprises a Defensive Responding Scale, which assesses the parents’ tendency to deny or minimize the difficulties associated with parenting. Overall, the sum of the scores obtained in the PSI-SF scales allows for an estimation of the level of Total Stress.

The second questionnaire used was the Eyberg Child Behavior Inventory (ECBI) [25], a 36-item measure administered to parents that assesses child behavior from a parent’s perspective. It includes an Intensity Scale, which measures the frequency of each behavioral issue and varies from 1 (never) to 7 (always), and a Problem Scale, that assesses how problematic a child’s behavior is in their parent’s opinion (for each item it is asked “Is this a problem for you?”). In the latter scale, each item allows to choose “Yes” or “No” as possible answers.

The third questionnaire used was the Maternal Attachment Inventory (MAI) [26], which is based on Attachment theory [27] and assesses maternal attachment towards the child. The MAI is composed of 26 items concerning actions or feelings associated with maternal affection towards the child that are rated on a scale from 1 (almost never) to 4 (almost always).

The questionnaires were of a self-report nature, but it is important to underline that some women of the detained group were unable to understand the meaning of some items fully, so it was necessary for the researcher to read and explain the questions to the participants. In the control group, women completed the surveys in a self-report form.

### 2.3. Data Analysis

In order to analyze the difference between the two parenting conditions (home detention and free living), analyses of variance (ANOVAs) were performed, with Group (detained vs. control) as the between-group factor, on the scores of parenting stress (PSI-SF), child behavior from the parent’s perspective (ECBI), and maternal attachment (MAI).

Analyses were conducted using the “Statistica8” software (StatSoft, Inc., 2007, Tulsa, OK, USA).

## 3. Results

The descriptive statistics of the participants were reported in Table 1. ANOVAs were performed on 12 women, divided into two groups (six detained and six control)**,** to assess the differences in parenting stress (PSI-SF), child behavior from the parent’s perspective (ECBI), and maternal attachment (MAI). The main effects of the detention situation were reported on the PSI-SF score for total stress, dysfunctional mother–child interaction, difficult child, and defensive responding, with the detained group reporting higher scores than the control group. There were no significant differences between the two groups for ECBI and MAI scores. The results of the questionnaires are summarized in Table 2.

## 4. Discussion

The aim of this preliminary report was to explore the differences between mothers living in detention through alternative measures with their children and mothers who are not detained regarding parenting stress, child behavior from the parent’s perspective, and maternal attachment.

As hypothesized, the main finding showed significant differences in the levels of parental distress between the detained mothers and the control ones. In particular, compared to the control mothers, mothers under alternative measures of detention reported higher levels of dissatisfaction concerning interactions with the child and higher disapproval of the child’s behavior. Moreover, mothers detained through alternative measures reported higher negative perceptions of the child’s capability for emotional regulation and a greater attitude toward denying or minimizing the difficulties associated with parenting. These findings suggest that mothers in alternative detention would experience higher levels of distress, as expected. Although the alternative measures of detention, such as Group Homes and home detention, are aimed at supporting parenting, in this preliminary report, the mothers in detention experienced a higher level of distress than those who are not in detention. This is in accordance with the current literature, as mothers in detention in other countries, such as in the US, were reported to display higher levels of stress compared to the general population [9,10,11,28]. Following the preliminary findings of the present report, it seems that the current measures of alternative detention in Italy do not sufficiently provide the parenting support necessary for protecting the well-being of both mother and child. Currently, there are no studies that compare motherhood in prison and motherhood in detention under alternative measures, such as in Group Homes and home detention. Through this comparison, it would be possible to fully understand if specific alternative measures could actually improve the quality of the relationships between detained mothers and children and their psychological well-being compared to conventional detention. Therefore, further studies should be planned to fill this important gap.

Despite the fact that children living in alternative measures facilities did not exhibit higher levels of dysfunctional behaviors compared to the control group, these results have likely been influenced by the limited sample size, and it might be possible to observe such differences when working with larger samples. This seems to be supported by a trend in the ECBI’s numbers, suggesting higher levels of dysfunctional behaviors among the detained group in comparison to the control group. While there are no studies in the current literature that concern behavioral functioning in children living in alternative measures facilities, it is known that children who live in prison with their mothers in Italy frequently suffer from sleep disorders, inappetence, apathy, and restlessness [8]. More broadly, it has been reported that children whose parents are involved in the criminal justice system are at increased risk of developing behavioral difficulties, such as engaging in delinquent behavior and reporting higher levels of externalizing problems, which can result in aggressive and antisocial behaviors [12,13,14,15]. These studies, while relevant, were conducted in the UK, USA, and Canada, where the criminal justice systems differ from the Italian one. This further underlines the necessity of studies that assess the impact that living in alternative detention facilities, such as Group Homes and home detention in Italy, can have on children’s behavioral functioning.

The analyses did not show any significant differences in maternal attachment between the two groups in the Maternal Attachment Inventory scores (MAI) [26] (Table 2). Nonetheless, it is possible that the results obtained by the detained group might have been slightly influenced by the mothers in order to obtain a juridical benefit regarding child custody or in the legal judgment of their detention by exhibiting a healthier functional pattern of maternal attachment. Supporting this, mothers living with or without their children while serving a sentence have been reported to experience a sense of inadequacy, failure, and shame concerning their parenting abilities [19]. Furthermore, the mother’s perception of her ability to perform her role in a socially acceptable way may shape her view of herself as a mother [9]. In this regard, the experience of imprisonment could create confusion about the maternal role and threaten one’s identity [9,29]. Therefore, it could be possible that mothers from the detained group may have slightly altered some scores in order to protect themselves and their image of themselves as mothers so that they could live with their child without disturbance or separation from any institution.

It is known that the adoption of the role and the identity of the mother starts with pregnancy and continues with early motherhood. This process could be affected by environmental precarity and personal adversity [30], suggesting that they might be impacted by incarceration. It is, therefore, possible that these multiple biasing factors have altered the test results.

Due to the lack of research concerning mother–child attachment in the context of detention through alternative measures, more studies are needed to comprehend how the imprisonment experience can affect maternal attachment and identity.

Overall, the study presented several limitations. Primarily, the size of the sample was small. Although 12 subjects can be sufficient to conduct an exploratory study on a topic that is scarcely investigated, data analyses are limited in generalizability, diversification, and impact. In addition, the age range of children (2–12 years) considered in the present study was quite broad and could potentially lead to discrepancies in the interpretation of the results, considering the specificity of the developmental stages that occur during these ten years. Further studies should monitor this variability to verify the effect of developmental stages on mother–child relationships in detained conditions.

Moreover, the questionnaires used to carry out this study were not adequate regarding the characteristics of the detained group of the present study, probably due to the gap concerning the language skill between the two groups. Indeed, in the detained sample, five out of six women were not able to complete the administration through the self-report mode because they did not fully understand the meaning of some items; therefore, the experimenter had to read and explain the questions to each one of them.

Previous studies suggested that the level of schooling and literacy is a very important factor affecting performance on tests in general, especially on cognitive tests [31]. However, as of today, there is a lack of translated and validated questionnaires for participants belonging to different ethnic and social minorities, such as those considered in the detained sample of the present study. Consequently, there is a strong need to adapt and validate specific questionnaires in order to obtain results comparable among different ethnicities and social minorities.

Another limitation of the present study was the social desirability bias that was presumably more present in the detained group.

The detained group can be defined as a niche within a niche since they were primarily incarcerated mothers, and secondly, five out of six women were of Roma ethnicity. As far as it has been possible to observe, detained women living with children in Italian prisons are mainly from the Roma community [32]. In Europe, the term “Roma” is often used to refer to a very wide range of different communities, such as the “Roma”, the “Gypsies”, the “Travellers”, the “Manouches”, the “Ashkali”, the “Sinti”, and the “Boyash” populations [33]. In Italy, people from Roma culture can no longer be described by the term “Gypsies”, as the word has become associated with racially disparaging epithets [34]. The Roma community has faced a long global history of genocide, exile, discrimination, and rejection by occidental societies over the years [34]. Moreover, Roma individuals have faced years of slavery [35], and unfortunately, there are multiple prejudices and stereotypes related to Roma culture. They are often accused of being “filthy” nomads, of stealing, and of not wanting to integrate with the “host” society [36]. Indeed, in modern society, the Roma community is often subject to racial discrimination, and this fact may intensify the difficulties of the scientific community to reach this population and deepen variables of interest [34].

Regarding the practical developments of this preliminary study, it should be noted that there is a strong social need to raise awareness on the topic. In fact, many people are not aware of the existing conditions of detained mothers living with their children and, more broadly, of their relationship dynamics and of the related social implications.

The inexistence of research tools validated for ethnic and social minorities contributes to fueling social and ideological distancing towards those minorities in contemporary society that could be filled by the scientific community.

## 5. Conclusions

In conclusion, the findings of the present preliminary report show that mothers living in alternative measures detention facilities are more stressed than mothers not in detention. In particular, when compared to control mothers, mothers detained in alternative measures facilities reported higher levels of dissatisfaction concerning interactions with the child and higher disapproval of the child’s behavior. Moreover, mothers detained in alternative measures facilities reported higher negative perceptions of the child’s capability for emotional regulation and a greater attitude toward denying or minimizing the difficulties associated with parenting. These findings suggest that mothers in detention experience higher levels of distress in general.

These findings suggested planning studies to verify the psychological impact of alternative detention on the well-being of mothers and children. It seems necessary to design intervention programs to support parenting in prisons, ICAMs, residential homes, and protected Group Homes [7,19]. The goal would be to reduce the risk of recidivism, support healthy motherhood, and increase the development of a stable mother–child relationship [19,28,37,38].

## Figures and Tables

**Table 1 ijerph-19-06000-t001:** Characteristics of participants.

	Detained Group (*n* = 6)	Control Group (*n* = 6)
Mother’s age (m ± sd)	34.0 ± 8.7	36.0 ± 7.0
Child’s age (m ± sd)	4.5 ± 2.2	4.5 ± 1.5
Nationality (*n*, %)	Bosnian (2, 33.2%); Italian (2, 33.2%); Croatian (1, 16.6%); Serbian (1, 16.6%)	Italian (6, 100%)
Level of education (*n*, %)	No education (1, 16.6%); Third grade level (1, 16.6%); Fifth grade level (1, 16.6%); Sixth grade level (2, 33.2%); High school (1, 16.6%)	High school (3, 50%); Bachelor’s degree (3, 50%)
Detention placement (*n*, %)	Home detention without child (1, 16.6%); Group Home “Leda Colombini” (4, 66.6%); Roma Camp (1, 16.6%)	N.A.
Presence of a second parent (*n*, %)	No (1, 8.3%); Yes (5, 91.6%)	Yes (6, 100%)
Presence of a partner (*n*, %)	Yes (5, 83,4%), No (1, 16.6%)	Yes (5, 83.4%); No (1, 16.6%)

**Table 2 ijerph-19-06000-t002:** ANOVA with the group factor (detained vs. controls) as the between-group factor on scores from the Parenting Stress Index (PSI), Eyberg Child Behavior Inventory (ECBI), Maternal Attachment Inventory (MAI).

	Detained (*n* = 6) m ± sd ^1^	Control (*n* = 6) m ± sd	F (1, 10)	*p*	Post-Hoc
PSI total stress	60.5 ± 10.5	45.7 ± 5.2	9.6	0.011	detained > control
PSI parental distress	28 ± 5.7	20.8 ± 5.7	4.7	0.054	
PSI parent-child dysfunctional interaction	24.5 ± 5.3	16.8 ± 1.8	11.1	0.007	detained > control
PSI difficult child	26.3 ± 4.1	20.2 ± 1.8	10.9	0.007	detained > control
PSI defensive responding	18.3 ± 3.4	12.2 ± 4	8.1	0.017	detained > control
ECBI intensity	107.5 ± 16.6	80.7 ± 26.5	4.4	0.062	
ECBI problem	5.8 ± 5.5	1.8 ± 2.6	2.6	0.138	
MAI total score	103.5 ± 1.2	103.5 ± 0.8	0.0	1.00	

^1^ Notes: m = mean; sd = standard deviation; *p* = *p*-value.

## Data Availability

The data presented in this study are available on reasonable request from the corresponding author. The data are not publicly available due to privacy reasons.

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
