# Peer review of "Motherhood in Alternative Detention Conditions: A Preliminary Case-Control Study"

_ijerph, 2022, doi:10.3390/ijerph19106000_

Round 1

Reviewer 1 Report

This is an important preliminary study assessing the parenting stress mothers experience in detention. It also highlights issues with child behavior and maternal attachment. The authors have incorporated sufficient background information with appropriate references. However, the last sentence of the introduction, “Results showed a higher level of stress in detained mothers,” gives away the critical information about the manuscript, and I feel it should be taken out from the introduction. The method section looks good. The results highlight that imprisoned mothers exhibit higher stress levels; interestingly, there was no significant difference in child behavior from the parent's perspective (ECBI) and maternal attachment (MAI), which, as the authors described, may be influenced by the mothers. The authors have also included several limitations of the study. Overall, this is a critical topic showing a need for better intervention programs to support imprisoned mothers.

Comments:

  1. One of the inclusion criteria was the child living with the mother for at least three years, so how is the child’s age 2-12?
  2. The exclusion criterion for both groups was the inability to speak and read Italian, so why did the researchers have to read the questions to the participant? Was it a reading issue or understanding the question?

Author Response

Response to Reviewer 1 Comments

Point 1: This is an important preliminary study assessing the parenting stress mothers experience in detention. It also highlights issues with child behavior and maternal attachment. The authors have incorporated sufficient background information with appropriate references. However, the last sentence of the introduction, “Results showed a higher level of stress in detained mothers,” gives away the critical information about the manuscript, and I feel it should be taken out from the introduction. The method section looks good. The results highlight that imprisoned mothers exhibit higher stress levels; interestingly, there was no significant difference in child behavior from the parent's perspective (ECBI) and maternal attachment (MAI), which, as the authors described, may be influenced by the mothers. The authors have also included several limitations of the study. Overall, this is a critical topic showing a need for better intervention programs to support imprisoned mothers.

Response 1: The authors thank the reviewer for her/his appreciations. Following the reviewer suggestion, the sentence was taken out from the introduction.

Point 2: Comments:

One of the inclusion criteria was the child living with the mother for at least three years, so how is the child’s age 2-12?

Response 2: The authors thank the reviewer for notice this error. The correct inclusion criterion was that the child had been living with the mother for at least two years. In the new version of the manuscript the correction was made.

Point 3: The exclusion criterion for both groups was the inability to speak and read Italian, so why did the researchers have to read the questions to the participant? Was it a reading issue or understanding the question?

Response 3: The authors thank the reviewer for notice this lack of clarity. During the enrollment some detained women were not able to fully understand the meaning of some items, so the researcher had to read and explain some questions to them.

In the new version of the “Materials” section, this issue was clarified as follows:

“Questionnaires were self-report, but it is important to underline that some women of the detained group were unable to fully understand the meaning of some items, so it was necessary for the researcher to read and explain the questions to the participants.”.

Moreover, in the new version of “Discussions” section the following sentence was revised as follows:

“Indeed, in the detained sample, five out of six women were not able to complete the administration through the self-report mode because they did not fully understand the meaning of some items, therefore the experimenter had to read and explain the questions to each one of them.”.

Reviewer 2 Report

See attached.

Author Response

Response to Reviewer 2 Comments

Point 1: The subject of this paper is important and interesting. However, it must be reviewed by a native English speaker once it is redrafted because I think it will help get some of the points across more clearly.

Response 1: The authors thank the reviewer for her/his appreciation. The new version of the manuscript was revised by a native English speaker.

Point 2: I have no objection to the way in which their study was conducted - in fact, I am not a social scientist, so I cannot comment on method fully. The only thing I might mention is that the age range of the children (2-12 years) in the sampled mothers seems to be quite broad and potentially could allow for much discrepancy.

Response 2: In order to address the limitation highlighted by the reviewer in the new version of the “Discussion” section the following sentences were added:

“In addition, the age range of children (2-12 years) considered in the present study was quite broad and could potentially lead to discrepancies in the interpretation of the results, considering the specificity of the developmental stages that occur during these 10 years. Further studies should monitor this variability to verify the effect of developmental stages on mother-child relationship in detained condition.”.

Point 3: I have some comments about the fundamental substantive basis of this piece. This piece compares women who have been detained in group homes with a control group of women who are just living at home with their children. But even that is not quite clear. Are the authors telling us that if a woman is convicted of a crime and sentenced to some form of imprisonment, they have three potential routes for detention: (1) detention and monitoring in their own homes if the conditions satisfy standard hygiene guidelines; (2) detention in prison; or (3) detention in one of the group homes? If so, I think that should be more clearly set out.

Response 3: The authors apologize for the lack of clarity. The present study aimed to compare a group of women who were detained through alternative measures with their children with a control group of women who were not detained.

In order to clarify this important point, in the new version of the “Introduction” section the following period was revised as follows:

“Therefore, the aim of this preliminary study was to explore the differences between mothers living in detention through alternative measures with their children and mothers who are not detained regarding parenting stress, child behavior from the parent’s perspective, and maternal attachment. Our first hypothesis was that mothers in detention through alternative measures would experience higher levels of stress compared to mothers not in detention. A second hypothesis was that children living with mothers in detention through alternative measures would report higher levels of difficult behavior because of the delicate living conditions. The third hypothesis was that maternal attachment would be higher in women in detention through alternative measures, presumably because of a greater protective instinct.”.

Moreover, according to the Italian law, the potential routes for detained mothers with their children are four: 1) conventional detention in prison, 2) a first alternative measure of detention which consists in specific prisons for mothers residing with the children (in Italy known as “Istituti a Custodia Attenuata per detenute Madri”, ICAM), 3) a second alternative measure of detention and monitoring in their own homes if the conditions satisfy standard hygiene guidelines (home detention), and 4) a third alternative measure of detention in a group home, which consist in spacious houses where several dyads of mothers and children can have their own room and space (Group Homes). 

In order to more clearly set out the possible alternative measures of detention, in the new version of the “Introduction” section the following sentences were revised as follows:

“Thus, in 2001, Law n. 40 made it possible for imprisoned mothers to serve their sentences through three alternative measures of detention, allocated in appropriate facilities [19]. The first alternative measure is a prison reserved for mothers who reside with children and which in Italy is known as "Istituti a Custodia Attenuata per detenute Madri" (ICAM). This alternative measure is a prison to all intents and purposes except for the establishment of environmental modifications to prevent children from recognizing the prison-setting they are housed in [20]. The second alternative measure is the home detention, applied for minor crimes, which allows the women to spend the sentence at their own home with their children, under electronic and police monitoring. When home detention is not possible due to adverse housing conditions, a third alternative measure is applied, the protected Group Homes. This kind of accommodation consists in spacious houses where some dyads of mothers and children can have their own room and space. While there are several controls carried by the police [21], the living environment more closely resembles that of a domestic home.”.

Point 4: My next question is whether there have been studies comparing women and children detained in prisons with women and children detained in the group homes. This comparison seems to be of more value because it seems evident that women and children who are able to live in their own homes would be subject to less stress and behavioral issues. It would be interesting to know whether the group homes actually provide a benefit that prisons do not provide. Of course they cannot redo the study now, but I think it would be helpful to acknowledge why they chose the control group to be women detained at their own homes and/or to reference any studies comparing detention in prisons with detention in the group homes.

Response 4: The authors thank the reviewer for this useful comment. At the best of our knowledge, there are no studies comparing women and children detained in prisons with women and children in detention through alternative measures, as group homes. Only few studies take into account the impact of alternative measures, but not in mothers’ populations.

Consequently, the purpose of the present study was to provide preliminary evidences about the impact of alternative measures of detention beginning with a comparison between a group of women in detention through alternative measures with a “control group” of mothers not detained. However, the authors understand the importance of comparing women and children detained in prisons with women and children in detention through alternative measures. Indeed, this comparison would be the most valuable strategy to understand if alternative measures, as group homes, actually provide a benefit that conventional detention in prison does not provide.

In order to acknowledge why the authors chose the control group to be women not detained, in the new version of the “Introduction” section the following sentences were revised as follows:

“Although alternative solutions to prison exist, in the literature there is a lack of data regarding their possible effects on the quality of mother-child relationships. It appears to be important to provide preliminary evidences about the impact of alternative measures of detention, by comparing a group of women in detention through alternative measures with a group of mothers not detained.”.

Moreover, in the new version of the “Discussion” section the following period was revised:

“Currently, there are not studies that compared motherhood in prison and motherhood in detention through alternative measures, such as Group Homes and home detention. Through this comparison it would be possible to fully understand if specific alternative measures could actually improve the quality of relationships between detained mothers and children and their psychological well-being compared to conventional detention. Therefore, further studies should be planned to fill this important gap.”.

Point 5: Finally, one small inquiry – in the paper the authors mention that there are just two group homes in Italy: Casa CIAO and Casa di Leda Colombini. However, when explaining on page 5 the recruitment of the detained group, the authors refer to three separate places, including a Roma camp. Can the Roma camp be compared to a Group Home? Is the Roma camp a Group Home? This is not clear and should be explained a bit more.

Response 5: In order to clarify what a Roma camp is the following sentence was revised as follows:

“The recruitment of women in Group Home detention took place at the “Leda Colombini” Group Home in Rome. The recruitment of women in home detention took place at the head office of the "A Roma, insieme Leda Colombini" Association and at the Roma camp of Castel Romano, which is a nomad camp located in Rome. Members of the Roma and Sinti communities are allowed to spend their house arrest inside the nomad camp.”.  

Round 2

Reviewer 2 Report

The authors have addressed all of my concerns and so I am happy with publishing it.